# Responsive Neurostimulation for Seizure Control: Current Status and Future Directions

**DOI:** 10.3390/biomedicines10112677

**Published:** 2022-10-23

**Authors:** Ujwal Boddeti, Darrian McAfee, Anas Khan, Muzna Bachani, Alexander Ksendzovsky

**Affiliations:** 1Department of Neurosurgery, University of Maryland School of Medicine, Baltimore, MD 21201, USA; 2Department of Neurosurgery, University of Alabama at Birmingham, Birmingham, AL 35294, USA

**Keywords:** seizure detection, seizure prediction, seizure controllability, seizure suppression by electrical stimulation, EEG processing for feedback control of seizure, mathematic modeling of EEG, system identification of EEG

## Abstract

Electrocorticography (ECoG) data are commonly obtained during drug-resistant epilepsy (DRE) workup, in which subdural grids and stereotaxic depth electrodes are placed on the cortex for weeks at a time, with the goal of elucidating seizure origination. ECoG data can also be recorded from neuromodulatory devices, such as responsive neurostimulation (RNS), which involves the placement of electrodes deep in the brain. Of the neuromodulatory devices, RNS is the first to use recorded ECoG data to direct the delivery of electrical stimulation in order to control seizures. In this review, we first introduced the clinical management for epilepsy, and discussed the steps from seizure onset to surgical intervention. We then reviewed studies discussing the emergence and therapeutic mechanism behind RNS, and discussed why RNS may be underperforming despite an improved seizure detection mechanism. We discussed the potential utility of incorporating machine learning techniques to improve seizure detection in RNS, and the necessity to change RNS targets for stimulation, in order to account for the network theory of epilepsy. We concluded by commenting on the current and future status of neuromodulation in managing epilepsy, and the role of predictive algorithms to improve outcomes.

## 1. Introduction

Epilepsy affects 70 million people worldwide [1]. It is characterized by spontaneous seizures which may occur in conjunction with other neurological, intellectual, or motor symptoms in the form of epilepsy syndromes [2]. Thirty percent of patients are refractory to medical therapy, which results in a significant impact on their quality of life. Thus, epilepsy imposes a significant health burden and financial strain on hospital systems worldwide, making this disease an important global health concern [1].

In conjunction with clinical features, clinicians can diagnose epilepsy and classify patients based on specific epilepsy subtypes, using diagnostic tools such as electroencephalography (EEG), video electroencephalogram (VEEG), computed tomography (CT), and/or magnetic resonance imaging (MRI) [3,4]. Broadly speaking, there are two main types of epilepsy: focal and generalized [3]. Of these two, the most common is focal epilepsy, in which seizures are confined to one hemisphere, as opposed to generalized epilepsy, which involves both hemispheres. Most focal epilepsies are caused by structural brain abnormalities, and can present on neuroimaging with atrophy, hyperintensity, or abnormal morphology. The most common type of focal epilepsy is temporal lobe epilepsy (TLE); most of these cases localize in the mesial temporal lobe structures (i.e., hippocampus, amygdala, and parahippocampal gyrus). Pathologically, TLE is most commonly associated with hippocampal sclerosis; however, other pathological lesions that are associated with focal DRE include long-term epilepsy-associated tumors, and malformation of cortical development [5]. Focal epilepsy can also arise outside the hippocampus, and involve regions such as the temporal neocortex, frontal lobe, occipital lobe, or the parietal lobe.

The first-line management for patients with epilepsy involves anti-seizure medications (ASMs). However, when at least two trials of ASMs fail to achieve seizure control, a surgical workup to identify the seizure onset zone (SOZ) is warranted [6,7]. This takes place in epilepsy centers that have the necessary tools for the tripartite surgical epilepsy workup [6]. Phase one of this workup involves noninvasive tools such as scalp EEG to localize seizure onset, and clinical evaluation to characterize seizure semiology [8]. In addition, MRI, positron emission tomography (PET), neuropsychiatric assessment, and the Wada test are used to localize language and memory [8]. However, the mainstay of this workup is the ability to obtain EEG recordings.

Before the first application of EEG in human recordings by Hans Berger in 1929, there was no way to quantify ongoing neuronal activity to help understand normal and pathologic functional states [9,10,11]. By 1934, EEG had helped characterize and differentiate normal human brain waves and seizure patterns in patients with brain tumors and epilepsy [10,11]. By measuring the local voltage fluctuations, clinicians and neuroscientists could identify physiologic and pathologic neuronal activity in superficial regions of the brain. EEGs are recorded from electrodes that are affixed to the scalp. Unfortunately, one flaw in these scalp EEGs is that they are limited in spatial resolution, as a result of recording signals through the skull and intermediate tissue [12]. This is where the advent of ECoG recordings changed how we measure brain waves and seizure patterns.

In 1934, the first use of intraoperative ECoG data by Foerester and Altenburger provided improved spatial resolution necessary to quantify electrical activity in both superficial and deep brain structures [13,14,15,16,17]. Subdural electrode grids and strips were used for direct recording from the cortex, while depth electrodes allowed for recording from deep structures. Analysis of ECoG data became a vital step in clinical decision-making for surgical resection or neuromodulation, and is the mainstay of phase two of the surgical epilepsy workup.

Phase two occurs if patients are candidates for surgery, and warrant further workup. During phase two, intracranial EEG (iEEG) studies take place with the placement of subdural electrodes, stereotactic electroencephalography (SEEG) electrodes, or a combination of the two. If a SOZ is identified, patients move on to phase three, which involves surgical intervention or neuromodulation of the epileptogenic focus (EF), in order to achieve seizure control. This most commonly involves open-surgical resection, but can also include magnetic resonance-guided laser-interstitial thermal therapy (MR-guided LITT), or neuromodulatory techniques such as responsive neurostimulation (RNS). Vagus nerve stimulation (VNS) and deep brain stimulation (DBS) are reserved for patients with multifocal epilepsy or generalized epilepsy, among other indications [12].

In this review, we first introduced surgical intervention for the treatment of DRE, discussing resection and neuromodulatory interventions available. Then, we discussed the emergence of therapeutic mechanisms behind RNS. Mainly, we aimed to contribute to a discussion on how the shift in understanding of epilepsy as a network disorder warrants a reevaluation in how we use RNS to improve seizure control. Finally, we discussed the application of machine learning to improve seizure detection in RNS, and the use of RNS in conjunction with resective surgery to achieve improved seizure control.

## 2. Surgical Intervention in Epilepsy

ECoG data collected during phase two of the surgical epilepsy workup is interpreted by epileptologists to identify the SOZ, in order to localize seizure activity and determine surgical options. This was first established by Penfield and Jasper, who were the first to institute iEEG as the mainstay for identifying the SOZ [18]. The most effective surgical intervention for DRE stands to be surgical resection, when possible. Other interventions can include MR-guided LITT, which is preferred when surgical resection carries a risk of high morbidity, or patient preference precludes a craniotomy [19]. MR-guided LITT involves the use of laser guided thermal energy to ablate the SOZ [19]. Removal of the epileptogenic zone was once thought of as a curative measure to eliminate predetermined foci involved in seizure onset. Temporal lobe resection is the most common resection procedure, and achieves seizure freedom in 64–85% of DRE patients with TLE [20,21]. However, when resective or ablative surgery is not an option, neuromodulation becomes an alternative therapeutic intervention. When there is not a single recognized SOZ, or it is located within highly functional brain regions, neuromodulatory interventions (i.e., DBS, VNS, RNS) can serve to disrupt seizure onset and spread, achieving seizure control and even seizure freedom in some cases [22,23].

## 3. Introduction to Neuromodulatory Therapies

Electrical stimulation of the cortex, a procedure that was pioneered by Penfield and Jasper, is commonly carried out both during and outside surgery, in order to map areas of neurological function [24]. This serves to help guide decision making with regards to anatomic regions that can be used to access deeper structures in the brain to target pathologies, such as brain tumors or EF, in the case of epilepsy. However, cortical stimulation can at times result in afterdischarges (AD), which are repetitive epileptiform discharges or aberrant spikes in electrical activity [25]. Lesser et al. observed that brief pulse stimuli (BPS) can counter these afterdischarges when administered prior to afterdischarge onset, and closer to the location of the afterdischarge onset [26,27]. The idea of using BPS to suppress epileptiform discharges underlies the idea of cortical stimulation to target EF, and is the premise behind the use of neuromodulatory devices, such as RNS, to target focal epilepsy [24].

Neuromodulatory techniques have been used as effective strategies for epilepsy and other neurological diseases. VNS was initially developed for epilepsy, but its use has been extended to the treatment of depression, migraines, Alzheimer’s Disease, obesity, and eating disorders such as bulimia nervosa [28,29]. For VNS, a device is implanted into the chest that allows for therapeutic modulation of the cervical truck of the left vagus nerve for patients with both focal and generalized seizures [30]. In epilepsy, the rate at which patients have been found to experience at least a 50% reduction in seizure frequency (also known as the 50% responder rate) from VNS is between 45–65% (Table 1) [31,32]. On the other hand, DBS was initially intended to treat patients with motor disorders, such as essential tremors and Parkinson’s Disease [33]. Later, studies showed that DBS, with implanted electrodes supplying a predetermined electrical stimulation to the anterior nucleus of the thalamus (ATN) and the hippocampus, improved seizure control in patients with epilepsy [30,34]. DBS has also been found to be beneficial in controlling electrographic sub-clinical seizures, when electrode leads are placed in the centromedian nucleus (CMN) of the thalamus in patients with Lennox–Gastaut syndrome [35]. Other controlled trials, carried out by groups such as Valentin et al., showed that DBS is safe and efficacious for the treatment of refractory generalized epilepsy [36]. The use of DBS in epilepsy has been found to have a 50% responder rate after one year of 43.4%, and up to 74% after seven years in long-term follow-up studies (Table 1) [37]. Both VNS and DBS have open-loop stimulation schedules which are independent of the underlying neuronal activity. In other words, the stimulation frequency and parameters do not change if a patient has a change in his/her seizure patterns on a given day. Therefore, efficacy could be improved through VNS and RNS devices that can modify stimulation parameters based on read brain activity.

## 4. Introduction to RNS

The RNS device is an intracranially implanted device that is used to achieve seizure control in patients with focal seizures (Figure 1) [38]. RNS was approved for the treatment of DRE in 2013 [38,39,40]. Specifically, the device consists of a neurostimulator that is attached to two leads, either of which can be NeuroPace depth leads or NeuroPace cortical strip leads. Each lead has four electrodes at the distal end that are implanted at the site of the SOZ or seizure spread (more recently), and four electrodes that are implanted at the proximal end attached to the neurostimulator [38]. Data from the RNS device can be recorded, and settings can be made using the programmer, remote monitor, and patient data management system (PDMS) database (Figure 2). The programmer allows the physician to change stimulation settings, and the remote monitor is a home-monitoring device that patients can use to upload ECoG data that are recorded by the RNS device to the PDMS database, which is a cloud storage of ECoG data, for physician review at a later time [38]. After implantation, the RNS device is set to passively record ECoG data without applying electrical stimulation for a period of time necessary, in order to define seizure neurophysiology, known as the programming epoch [41]. After this time, stimulation settings are programmed into the RNS device, after which it is capable of delivering electrical stimuli in response to detected epileptiform activity [41]. The recommended initial responsive therapy settings are as follows: frequency: 200 Hz; burst duration: 100 ms; current: set to achieve a charge density = 0.5 μC/cm^2^; pulse width: 160 μsec [42]. The main metric that is adjusted when changing the stimulation parameters is charge density, which is increased by 0.5 μC/cm^2^ at each programming visit, should the response at the current charge density level be unsatisfactory [42]. The pivotal difference between RNS and other neuromodulatory devices is that RNS is a closed-loop system with sensing capabilities [34]. This means that the neurostimulator that is implanted delivers direct electrical stimuli in response to the detection of specific patterns of electrographic activity that have been predetermined by the physician to be epileptiform activity [38]. Additionally, the RNS device records and stores ECoG data for physician review, allowing for access to, and analysis of, long-term ECoG data [40]. The ECoG data are collected as a bipolar differential between neighboring electrodes on a lead, and are sampled at 250 Hz [38]. This enables physicians to access ECoG activity immediately before and immediately after specific events. Events that trigger the recording of EcoG data include: (1) detection of epileptiform activity, (2) responsive stimulation, (3) long duration of detection event, (4) magnet swipe by patient, (5) saturation (high-amplitude activity), or (6) detection of noise (60 Hz line noise) [38]. The long-term efficacy of RNS is comparable to DBS, with RNS achieving one-year, two-year, and five-year 50% responder rates of 44%, 55%, and 50–61%, respectively (Table 1) [30,43].

## 5. RNS for Focal Epilepsy

Three clinical trials ultimately led to the U.S. Food and Drug Administration’s (FDA) approval of the RNS device to treat focal epilepsy. The first was a feasibility study that confirmed the safety of the device, allowing for a second clinical trial, a pivotal study. The pivotal study, which was a two-year multicenter, double-blinded, randomized controlled trial, consisted of 191 patients who underwent implantation of the RNS device [44]. Patients who were implanted were randomized to either the active stimulation group or the sham group. For the duration of the pivotal study, the active stimulation group demonstrated a 37.9% reduction in mean seizure frequency, compared to only 17.3% reduction in the sham group [44]. Further out from implantation, the active group continued to show improved seizure control, with the median percent seizure reduction being 44% at one year, and 53% at two years [38].

After completion of the feasibility and pivotal studies, an open-label long-term treatment (LTT) study continued to follow patients who were implanted with the RNS device, either in the feasibility study or in the pivotal study, and looked at outcomes for an additional 7 years [44]. The median seizure reduction rate at 9 years post-implantation was found to be 75% [44,45]. Outcomes in the LTT study were broken down by location by Geller et al., who explored outcomes in mesial temporal lobe epilepsy (MTLE), and Jobst et al., who explored outcomes in neocortical epilepsy [24]. Jobst et al. identified that RNS achieved improved seizure control in neocortical epilepsy when patients had a structural lesion relative to those who did not (median 77% reduction vs. 45% reduction, respectively), suggesting that RNS device lead placement may be more important to achieving better seizure control when targeting neocortical epilepsy targets [46]. This was not found to be the case in MTLE [47].

The true underlying mechanism behind how RNS exerts its therapeutic effect is unknown. To date, the most widely accepted understanding follows Lesser et al.’s observations that BPS applied to cortical locations closer to afterdischarge onset can control the aberrant electrical activity [41]. In other words, the idea behind RNS to identify regions of epileptic activity, and then provide neutralizing or disruptive activity at the site (direct inhibition/suppression of epileptic activity), in order to restore normal function [38], [41]. However, more recent studies suggest that the therapeutic effect of RNS is likely to be driven by modulation of epileptic networks.

## 6. RNS Modulation of Epileptic Networks

In order to better understand the more recent findings behind the therapeutic mechanism underlying how RNS works, it is important to discuss the recent shift in thinking of epilepsy as a network disorder.

### 6.1. Epilepsy as a Network Disorder

Until relatively recently, epilepsy has been thought of as a focal disease; however, recent studies suggest that epilepsy is a disorder of a distributed epileptogenic network [48,49,50]. In 1951, Bailey and Gibbs wrote that “surgical resection of focal seizure activity was comparable to eradicating a tumor,” delineating the former understanding of surgical resection of EF as curative [51]; however, post-surgery outcomes show that 42–63% of patients continue to have seizures within one year of surgery, dispelling this former train of thought [48]. In order to determine the reason for seizure recurrence, recent clinical studies have shown that the resection or modulation of nodes that are involved in early seizure spread from the SOZ may result in significantly improved seizure control, post-surgical resection, as opposed to resection of nodes involved in later seizure spread. Specifically, in a cohort of patients with TLE who had resection of EF involved in early seizure spread (<10 s) from seizure onset, had an approximately 90% reduction in seizures post-resection [48]. Additionally, these findings were further supported by studies that showed that resection and/or neuromodulation of nodes involved in high interictal-connectivity led to enhanced post-operative freedom [52]. For example, Sisterson et al. showed in a retrospective analysis that RNS with electrodes implanted in the CMN of the thalamus, an important node in refractory generalized epilepsy, resulted in a significant reduction in seizure frequency (75–99%) and severity [53]. These findings suggest that (1) epilepsy is a network disorder, and that (2) EFs identified in the epilepsy monitoring unit (EMU) do not correlate with targets that achieve maximal seizure freedom [48].

### 6.2. Modulation of Epileptic Network

Given the former understanding of epilepsy as a seizure-focus disorder, it was originally thought that RNS worked by disrupting epileptiform activity at implanted sites that were predetermined to be EFs. However, recent studies suggest additional therapeutic potential for RNS as a means to disrupt network activity.

In 2019, Kokkinos et al. performed a retrospective review of ECoG recordings from 11 patients with focal epilepsy who were implanted with RNS devices [41]. Specifically, they looked at ECoG data in the time–frequency domain, and identified two major categories of effects from electrical stimulation: direct effects and indirect effects. Direct effects were characterized as time or frequency changes that occurred in the immediate period (<5 s) after a responding stimulation was applied by the RNS device. Indirect effects were characterized as changes in time or frequency that occurred at least 27 s after a previous stimulation, and at least 11 s before the next. Direct effects included (1) ictal inhibition, where RNS stimulation resulted in the ECoG data returning to the interictal level within 5 s; and (2) frequency modulation of the ECoG data, where there were changes in the active frequency bands in ECoG data recorded within 5 s of a stimulation event. Indirect effects included (1) spontaneous ictal inhibition, in which seizure activity resolved spontaneously in the absence of electrical stimulation by the RNS device; (2) spontaneous frequency modulation of the ECoG data; (3) fragmentation, where seizure activity was spontaneously disrupted by periods of normal interictal activity; and (4) spontaneous decrease in ictal duration. Seizure outcomes were measured using the extended personal impact of epilepsy scale questionnaire, subjective measures of seizure frequency, severity, and duration, and clinically determined Engel scale classes [41]. The authors demonstrated that the odds ratio (OR) for indirect modulatory effects was significant for the outcome measures of seizure frequency, severity, and duration (seizure occurrence frequency: OR = infinity, P = 0.005; seizure severity: OR = infinity, P = 0.007; and seizure duration: OR = 28.0, P = 0.03). In contrast, the OR for direct modulatory effects was not significant for any measure of seizure activity (seizure occurrence frequency: OR = 0.67, P > 0.99; seizure severity: OR = 0.0, P = 0.10; and seizure duration: OR = 0.25, P = 0.56). Thus, Kokkinos et al. concluded that improved clinical outcomes that were seen in RNS patients are likely attributed to the indirect effects of RNS, as opposed to direct suppression of focal epileptiform activity [41].

However, more recent studies are further suggesting that this indirect therapeutic effect of RNS is a result of disruption of epileptic network activity. In 2022, Fan et al. conducted a retrospective study of 31 patients with implanted RNS devices [51]. Specifically, they characterized the functional connectivity of these patients’ electrographic activity using resting-state magnetoencephalogram (MEG) data that were collected prior to RNS implantation, with the goal of determining if baseline functional connectivity can help predict clinical outcomes of RNS. Fan et al. looked at functional connectivity across various spatial scales, including global, hemispheric, and lobar, after spectral decomposition of patient ECoG data obtained from RNS. Fan et al. demonstrated that increased baseline functional network connectivity was associated with improved clinical outcomes, as measured by percent change in self-reported seizure frequency in the most recent year of clinic visits, compared to prior to RNS device implantation [54]. More specifically, they identified that increased global functional connectivity in the alpha frequency band was correlated with seizure frequency reduction (P = 0.010). Additionally, global functional connectivity was also more strongly predictive of responder status, compared to hemispheric functional connectivity. A similar study was carried out by Charlebois et al., who carried out a retrospective study of 22 patients with MTLE who were treated with RNS. Given that studies suggest MTLE is a network disorder that involves structures such as the amygdala, hippocampus, entorhinal cortex, cingulate cortex, thalamus, and hypothalamus, Charlebois et al. aimed to show that network modulation played a role in seizure reduction in MTLE, as opposed to stimulation location [55]. From their analysis, Charlebois et al. showed that the stimulation of regions connected to the medial prefrontal cortex, ipsilateral anterior cingulate, and contralateral precuneus, was predictive of seizure reduction, in comparison to the volume of tissue activation (VTA) location. VTA refers to the collection of anatomical locations that show the spread in electrical field from stimulating leads [56]. In other words, VTA location is representative of the direct impact of stimulation. These results are concordant with the increased connectivity to the posterior cingulate, medial prefrontal cortex, and precuneus seen in DRE patients who responded to DBS at the ATN, as shown by Middlebrooks et al. [57] (Table 2).

In summary, studies by Fan et al. and Charlebois et al. suggest that baseline functional connectivity and stimulation of specific networks inform the clinical response to RNS. Ultimately, this suggests that the indirect therapeutic effects of RNS observed by Kokkinos et al. are likely due to the disruption of epileptic network activity, further supporting the network theory of epilepsy.

## 7. Improving Seizure Prediction and Control

Given our most recent understanding behind how RNS exerts its therapeutic effect, we can use this information to better guide surgical decision-making and inform RNS lead placement to better address highly-connected nodes. However, additionally, the seizure detection abilities of RNS can be further improved utilizing the large-scale ECoG data it offers. We will now discuss how RNS offers a form of long-term ambulatory ECoG data, how RNS can potentially better detect seizures using machine learning algorithms, and how RNS lead placement can be modified on the basis of our understanding of epilepsy as a network disorder.

### 7.1. Improving Seizure Prediction

Given that the basis for the function of RNS is seizure detection, it is imperative that the device successfully (1) identifies seizures with high accuracy and precision, and (2) detects these seizures early [58]. Currently, RNS detects seizures using three tools: the line detection tool, the area detection tool, and the bandpass detection tool. The line detection tool identifies changes in frequency and amplitude of the ECoG signal. The area detection tool identifies changes in total signal energy, without accounting for frequency. Finally, the bandpass detection tool serves as a frequency filter, and is used to detect activity within specific frequency bands (i.e., theta, alpha, beta, and gamma) [38].

Since RNS uses a template signal that is predetermined and programmed by epileptologists to be the trigger for stimulation, the system may fail to detect some seizures. This is where machine learning can be incorporated, using training periods at regular intervals in order to maximize effectiveness. This is an improvement over using one rigid template that never adapts, and has little room for error. For example, Lee et al. showed that a principal component analysis (PCA) frequency-based algorithm improved early seizure detection in a pilocarpine-induced epilepsy rat model [58]. PCA is a computational tool that allows for the identification of components of a signal that most contribute to the uniqueness of the signal; this enables the parsing of the most important aspects of ECoG signals that immediately precede epileptiform activity. Specifically, Lee et al. demonstrated that PCA was able to identify frequency-based features that were able to predict seizures more accurately from an early ECoG recording, in comparison to the ECoG data obtained from the entire duration of the seizure [58]. Studies such as these demonstrate the ability to accurately detect seizures by applying predictive algorithms to early ECoG data collected at the beginning of a seizure. However, more specifically, recent studies that attempted to train machine learning algorithms using the ECoG data obtained from the RNS system have been completed.

In 2021, Yueqiu et al. demonstrated the possibility of predicting seizure frequency by training five different machine learning algorithms with interictal frequency domain ECoG data obtained from five different patients implanted with RNS devices (Table 3) [34]. Specifically, their group compared performances (as measured by areas under the curve, AUC) across the following machine learning algorithms: (1) support vector machine (SVM), (2) logistic regression, (3) decision tree, (4) random forest, and (5) gradient boosting [34]. Yueqiu et al. trained each algorithm using 80% of a patient’s ECoG data, and then tested its performance against the remaining 20% of ECoG data. From their analyses, Yuequi et al. identified that high gamma power during the interictal period was predictive of high seizure-frequency epochs, and that overall, the best performing machine learning algorithms were SVM and gradient boosting. However, their group also showed that the best performing algorithm was not consistent from patient to patient.

In addition to using interictal ECoG data from RNS to predict seizure frequency, groups such as Constantino et al. explored whether machine learning algorithms could detect RNS-derived ictal patterns with an accuracy comparable to that of epileptologists (Table 3) [59]. Specifically, Constantino et al. trained a convolutional neural network (CNN) model to distinguish ictal activity from non-ictal activity [59]. A CNN model is a specific type of deep learning model; it is considered the model of choice when working with multiple array data, such as EEG data [59]. Constantino et al.’s group was able to show that with a large training set of RNS-derived ECoG data, a CNN model could detect seizures with an accuracy that was similar to that of expert epileptologists [59].

Although in recent years, machine learning algorithms have been applied to study ECoG data obtained from RNS, they have yet to be implemented to aid in advanced seizure detection. This is where we foresee that large-scale RNS-derived ECoG data can guide future studies that inform the adaptation of machine learning algorithms, in order to better achieve seizure control. For example, a theoretical CNN model could be trained to use time domain and spectral features in the interictal and pre-ictal period to predict seizure onset and seizure duration. Yueqiu et al. suggested that this information could then guide stimulation programming, both in the interictal period (akin to dampening activity that is “fueling the fire” of an upcoming seizure, a probable mechanism behind RNS’s therapeutic function in the first place, as discussed above (see Section 6.2)) and the immediate preictal period, detecting and controlling seizures that may potentially be missed by the RNS system.

However, there are still several questions that need to be answered in this realm. As Yuequi et al. showed, different machine learning algorithms perform better in different patients, which begs the question: what are the implications of this in RNS programming? Additionally, what stimulation parameters should algorithms adjust to over time, in order to improve seizure control? These are the pivotal questions that can be investigated, once we begin to explore the application of machine learning algorithms using RNS-derived ECoG data specifically to improve seizure detection [60].

### 7.2. Improving Seizure Control

RNS currently targets 1–2 foci in order to treat seizures, in approaching epilepsy as a focus-based disorder. However, as described above, studies are now showing that neuromodulatory devices exert their therapeutic effects through network disruption, not focus disruption [54]. Using this information, we can inform RNS target placement to address key nodes in the epileptogenic network, allowing for improved seizure control.

As discussed above (see Section 6.1), the specific selection of nodes involved in epileptic activity that are targeted has a significant impact on postoperative outcomes. Keeping this in mind, it is imperative that RNS lead placement be directed by the identification of nodes that play an integral role in the pathologic seizure network that underlies DRE. This can be accomplished in manner that is similar to how Andrews et al. stratified patients on the basis of the timing of seizure spread [48]. However, other connectivity metrics can also be implemented. For example, using ECoG data obtained from EMU recordings, functional connectivity analyses can be run to determine the most important targets for network disruption. Metrics, such as coherence, cross-correlation, or mutual information, can be used to identify targets that would otherwise not be identified by epileptologists in the EMU [61].

In addition to changing RNS targets, surgical epilepsy outcomes can also be improved using RNS as an adjunctive therapy to surgical resection. Although resection and neuromodulation are options that are available for patients with DRE, patients with whom resection surgery alone cannot address the ictal onset zone (i.e., multifocal epilepsy, eloquent areas) are deemed to be poor surgical candidates, and have minimal therapeutic options. However, recent studies are showing that these patients can benefit from a combination of resective surgery supplemented with RNS. One of the foremost studies that showed this explored a cohort of ten patients with multifocal ictal onset at the University of California Irvine (UCI), from 2015 to 2019, who underwent resection and RNS device placement [62]. Outcomes showed a significant decrease in seizure frequency (average 81% +/− 9) at the six month follow-up [62].

Using graph theory metrics to target RNS lead placement based on the most highly connected foci, as opposed to using traditionally determined EFs and RNS in addition to resective surgery, RNS can serve to treat epilepsy in a fashion concordant with the network-based pathomechanism of the neurological disorder, in order to achieve improved seizure control [63].

### 7.3. Ambulatory ECoG Data

In order to enhance RNS’s ability to improve seizure prediction, it is important to have a sufficiently large data set of ECoG recordings to train machine learning algorithms, as discussed above (see Section 7.1). This is where RNS’s ability to provide access to ambulatory ECoG recordings comes into play. Ambulatory ECoG data are important, clinically, because they provide objective information about the activity before and during a seizure that the patient may not remember having, including nocturnal seizures, consequently bypassing the limitations of seizure diaries [64]. Traditional intracranial ECoG data collected during phase two of the surgical epilepsy workup are obtained in the controlled setting of the EMU. However, there are limitations to traditionally obtained ECoG data. Firstly, because the electrodes are wired to the recording device, patient movement is restricted, resulting in a relatively stationary state during observation. Therefore, the diagnostic abilities and seizure frequency are not necessarily synonymous with what a patient experiences when carrying out their normal day. For this reason, the data collected during the few weeks in the EMU are likely an under-sampling of the patient’s natural seizure activity [65]. Furthermore, observations over time have shown that there exist cycles of seizures in patients with epilepsy in which interictal epileptiform activity (IEA) fluctuates cyclically [65]. As such, sampling of IEA during acute EEG or iEEG measurements in the EMU largely miss the IEA outside the current cycle being measured, further suggesting that ECoG data from the EMU are significantly limited in their diagnostic value [65]. With RNS, instead of relying on spontaneous seizures in the EMU, epileptologists can analyze months to years of natural epileptic data to provide more precise modeling of ECoG activity that precedes seizures, and make more informed medical recommendations to the patient [66]. Moreover, during this time, a patient is weaned off ASMs. When a patient ceases to take ASMs, their seizure profile changes in the EMU compared to that at home [67,68,69]. Therefore, the seizure activity that is recorded in the EMU may not be reflective of the baseline nature of the disease. Lastly, long-term patient ECoG data allow for patient-specific modeling, as well as for a further understanding of normal and pathologic brain activity. However, it is important to note that along with the added benefits of ambulatory ECoG data obtained from RNS, they are limited by spatial resolution, as there are only four electrodes per location being recorded from RNS. Additionally, RNS has only one battery, as opposed to ECoG data recorded from the EMU which use many electrodes, each with their own power source. In conclusion, RNS provides a superior method of obtaining ECoG recordings that can then be used to train machine learning algorithms to ultimately improve seizure detection and seizure control.

## 8. Conclusions

Outcomes in epilepsy surgery have been stagnant over the past two decades, and the mechanisms by which neuromodulation reduces seizure frequency are not clearly understood [41]. With growing consideration for epilepsy as a network disorder, instead of as a seizure-focus disorder, an in-depth understanding of the changes in network connectivity is pivotal in the care of EFs in high-functioning brain regions. For patients with DRE, more effective tools for seizure detection, seizure prediction, and network connectivity are needed in order to improve seizure reduction rates. Until these factors are addressed, patients will continue to have a detrimental quality of life. RNS has the capabilities to tackle these concerns.

Given the closed-loop nature of RNS, RNS has immense potential, not only in clinical care, but also in the mechanistic understanding of epilepsy. RNS has the capabilities to obtain and store long-term ambulatory ECoG data; these will provide clinicians and neuroscientists with the best guide to understanding and making significant segues in clinical decision-making. Regarding RNS’s therapeutic stimulation, while the effects are not yet fully understood, previous research suggests that its indirect effects, which correlate more toward clinical outcomes, are concordant with the network theory of epilepsy [70]. A shift toward addressing highly functional network nodes, instead of simply the regions of strongest epileptic activity, will be the next step in neuromodulatory therapy. Nonetheless, more understanding of epilepsy networks is necessary to achieve this.

More progress in RNS is necessary, before it reaches optimal efficacy rates. Firstly, RNS devices will require increased storage capabilities or periodic backups, in order to store the ECoG data they continuously obtain [64]. Secondly, ECoG data detection algorithms need to be adjusted, in order to consider highly connected network nodes instead of just one or two EFs. Lastly, the integration of machine learning into RNS workup and stimulation will allow for more personalized therapy through improved seizure prediction. Machine learning algorithms of the ECoG data will also help distinguish which patients will be more responsive to RNS. Overall, there is immense potential for RNS in the future treatment of epilepsy as we increase our understanding of network epilepsy and stimulation.

## Figures and Tables

**Figure 1 biomedicines-10-02677-f001:**
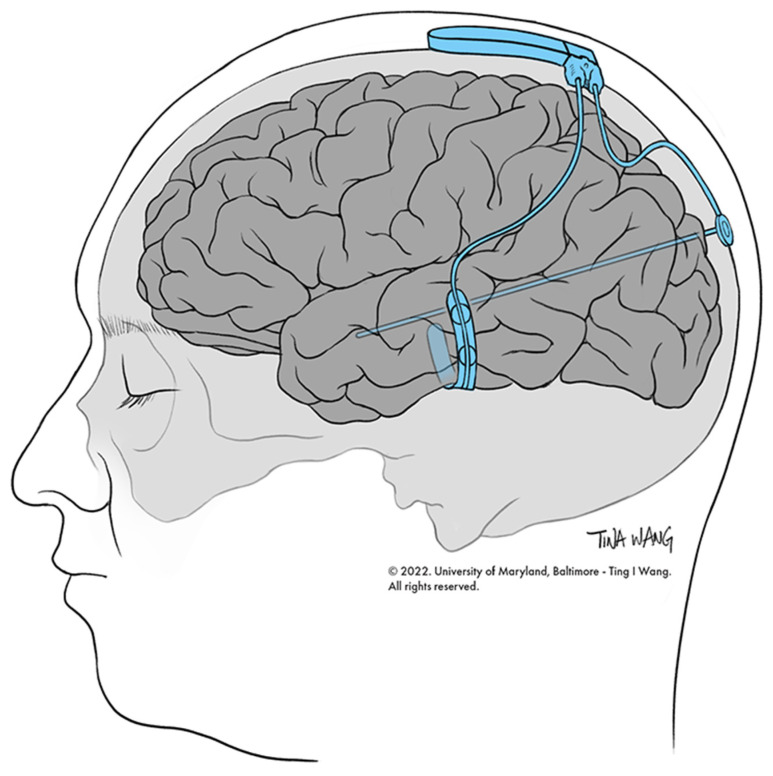
RNS system. Representation of implanted RNS system.

**Figure 2 biomedicines-10-02677-f002:**
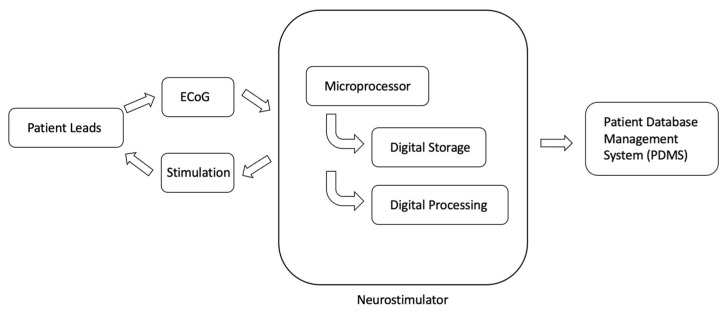
RNS system workflow. Representation of information flow from RNS patient leads, the RNS neurostimulator, and the PDMS.

**Table 1 biomedicines-10-02677-t001:** Neuromodulatory interventions 50% responder rates. Comparison of 50% responder rates across the following neuromodulatory interventions for the treatment of epilepsy: RNS, VNS, and DBS.

Neuromodulatory Intervention	50% Responder Rate
RNS	1 year: 44% 2 years: 55% 5 years: 50–61% [37]
VNS	1 year: 45–65% [31,32]
DBS	1 year: 43.4% 2 years: 53.7% 5 years: 67.8% 7 years: 74% [37]

**Table 2 biomedicines-10-02677-t002:** Studies suggesting that RNS works by seizure network disruption. Summary of the major recent studies that establish that the therapeutic mechanism behind RNS is likely epileptic network disruption and modulation.

Study	Authors and Year	Study Type and Number of Patients	Main Points
Association of Closed-Loop Brain Stimulation Neurophysiological Features With Seizure Control Among Patients With Focal Epilepsy	Kokkinos et al., 2019 [41]	Retrospective study, 11 patients	- The effects of RNS were separated into direct and indirect effects. - Indirect effects strongly correlated with clinical outcomes. - This suggests that therapeutic benefit of RNS comes from modulation of the seizure network as opposed to acute disruption of seizure events.
Network connectivity predicts effectiveness of responsive neurostimulation in focal epilepsy	Fan et al., 2022 [54]	Retrospective study, 31 patients	- Reduction in seizure frequency and responder status more strongly predicted by global functional connectivity, specifically in the alpha frequency band.
Patient-specific structural connectivity informs outcomes of responsive neurostimulation for temporal lobe epilepsy	Charlebois et al., 2022 [55]	Retrospective study, 22 patients	- Stimulation of regions connected to the medial prefrontal cortex, ipsilateral anterior cingulate, and contralateral precuneus in MTLE patients was predictive of seizure reduction, in comparison to the volume of tissue activation (VTA) location.

**Table 3 biomedicines-10-02677-t003:** Studies Applying Machine Learning Algorithms to RNS-Derived ECoG Data. Major recent studies that have used machine learning algorithms in the context of RNS-derived ECoG data to predict seizure frequency and ictal periods.

Study	Authors and Year	Study Type and Number of Patients	Machine Learning Algorithms Used	Main Points
Machine Learning to Classify Relative Seizure Frequency From Chronic Electrocorticography	Yueqiu et al., 2021	Retrospective study, 5 patients	- Support Vector Machine (SVM) - Logistic Regression - Decision Tree - Random Forest - Gradient Boosting	- High gamma power during the interictal period was predictive of high seizure-frequency epochs. - Best performing machine learning algorithms were SVM and Gradient Boosting. - Best performing algorithm was not consistent from patient to patient.
Expert-Level Intracranial Electroencephalogram Ictal Pattern Detection by a Deep Learning Neural Network	Constantino et al., 2021	Retrospective study, 22 patients	- Convolutional Neural Network (CNN) (Deep Learning Model)	- With a large training set of RNS-derived ECoG data, a CNN model can detect seizures at an accuracy similar to that of expert epileptologists.

## Data Availability

Not applicable.

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
