# Peer review of "Responsive Neurostimulation for Seizure Control: Current Status and Future Directions"

_biomedicines, 2022, doi:10.3390/biomedicines10112677_

Round 1

Reviewer 1 Report (Previous Reviewer 3)

This study reviews and summarizes benefits from neuromodulation devices such as responsive neurostimulation (RNS) for epileptic seizure management.

Strengths: The authors provide in depth analysis of RNS, its working principles, challenges to acceptance, the use of machine learning techniques to improve predictive outcomes, and the epileptogenic network view of seizure management.

Weaknesses: there is little to say in terms of weakness of this study. The authors have implemented the reviewer's suggestions.

Author Response

Reviewer 2 Report (Previous Reviewer 2)

I recommend the manuscript for publication.

Author Response

Reviewer 3 Report (Previous Reviewer 1)

The text material looks great. Recommend including at least 1 figure illustrating the RNS system and another showing PDMS workflow

Author Response

This manuscript is a resubmission of an earlier submission. The following is a list of the peer review reports and author responses from that submission.

Round 1

Reviewer 1 Report

Thanks for the giving me the opportunity to review this excellent work. Your review provided a concise yet comprehensive review on the role of responsive neurostimulation in management of epilepsy. But here are a few comments:

1- The manuscript lacks any figures which is very unusual in a neurostimulation paper covering both surgical and programming aspects of responsive neurostimulation. Suggest at least 2 figures one demonstrating the device and another showing PDMS Ecog or other views.

2- Paragraph 4 line 1 " Before the discovery of EEG in 1920s": This statement is not accurate. As the discovery of EEG in animals by Richard Caton in the 1800s preceded that. The first recording in humans by Hans Burger occurred in 1920s though. See  http://dx.doi.org/10.1136/jnnp.74.1.9 and :https://www.ncbi.nlm.nih.gov/books/NBK390348/

3- Paragraph 5 line 4: "In the 1970s, the use of electrocorticography provided spatial... etc.." The use of Ecog either through superficial recording obtained intraoperatively by the Pennfield group or via SEEG methodology by the Talairach and Bancaud group preceded that. See https://doi.org/10.1016/j.seizure.2016.04.006    

for more details.

3- Under the heading: "Introduction to Neuromodulation therapies": Paragraph 2 line 13-15" : It was mentioned that " DBS has been found  to beneficial in controlling seizures when electrode leads are placed in the centromedian nucleus of the thalamus". Dalic's paper was cited. Please refer to Dalic's paper. CM stimulation was superior to sham stimulation in controlling electrographic sub-clinical seizures but no statistically significant effect was found with regards to clinical seizures. Other open label studies by Velasco et al, Valentin et al , Cukiert et al and others have show beneficial effect of CM stimulation but it is important to note that the randomized controlled study by Dalic et al. demonstrated benefit in controlling electrographic but not clinical seizures. Also that statement is grammatically incorrect. Would add the word " be" before "beneficial"

4- Same paragraph line 18: It was stated that " Unfortunately both VNS and DBS have open-loop stimulation". Recommend omitting the word "unfortunately" unless you provide evidence that closed loop stimulation is superior to open loop stimulation in epilepsy which to my knowledge is not yet available as median seizure reduction rates with DBS are similar to RNS and small series of chronic sub-threshold cortical stimulation (CSCS) have shown similar benefit as well.

5- Under the heading" RNS for focal epilepsy" Paragraph 2 :  " The third clinical trial was an open-label... etc.." Not sure if I would label that as an independent clinical trial if it just included long term follow up of patients in the original pivotal and feasibility trials. Consider rephrasing that statement.

6-Under the sub-heading " Epilepsy as a network disorder" : Recommend adding data on role of RNS stimulation of thalamic targets including in generalized epilepsy. See 10.1136/jnnp-2021-327512

7- Finally, would recommend briefly discussing recommended initial programming settings and the recommended initial programming changes which are available in Neuropace's programming manual.

-My recommendation is accepting the manuscript after these revisions are completed. 

Reviewer 2 Report

The authors present the article entitled “Responsive Neurostimulation for Seizure Control: Current Status and Future Directions”.

This paper presents a review of the clinical management for epilepsy and discusses the steps from seizure onset to surgical intervention, reviewed studies discussing the emergence and therapeutic mechanism behind RNS, and discusses why RNS may be underperforming despite an improved seizure detection mechanism. 

The article presents the following concerns:

  • Introduction section must be synthesized. What are the contributions of the review? What is the objective of the review? I suggest describing the background concisely.  

  • Section 3: Please add a comparative table that includes the variables used in the literature. What are the perspective

  • The manuscript must present a discussion section that makes a deep comparison between the machine learning algorithms reported in the state of the art.

  • The references are split several times as refs 1 and 3. Idem for 10-11.

  • I suggest the authors provide a theoretical model of machine learning techniques in responsive neurostimulation for seizure control that could improve the reported ones in the literature. 

  • Line 72, regarding the growing of EEG signals for achieving outstanding result in healt can be justified with the following references: Impact of eeg parameters detecting dementia diseases: a systematic review; Cortical activity at baseline and during light stimulation in patients with strabismus and amblyopia; A new approach for motor imagery classification based on sorted blind source separation, continuous wavelet transform, and convolutional neural network.

  • The text must be written in the 3rd person or passive voice.

  • Add hyperlinks to tables, figures, and references.

  • Recommend making a little introduction between points 6 and 6.1, 7 and 7.1

  • Please add a nomenclature table to define variables and acronyms.

The following misspelling should be checked:

  1. line 83: The article “an” may be incorrect. Consider changing it to agree with the beginning sound of the following word “SOZ”.

  2. line 137: It appears that “resulted in” may be unnecessary in this sentence. Consider removing it. 

  3. line 138: It seems that you are missing a verb in this sentence “been found to beneficial…” consider adding it: “be beneficial”

  4. line 185: The phrase “in comparison” may be wordy. Consider Changing the wording by “compared”. 

  5. line 206: The phrase “are suggesting” may be wordy. Consider changing the wording by “suggest”. 

  6. line 388: Apostrophes must be avoided: “it´s”

Reviewer 3 Report

This study reviews and summarizes benefits from neuromodulation devices such as responsive neurostimulation (RNS) for epileptic seizure management.

Strengths: The authors provide in depth analysis of RNS, its working principles, challenges to acceptance, the use of machine learning techniques to improve predictive outcomes, and the epileptogenic network view of seizure management.

Weaknesses: there is little to say in terms of weakness of this study, except that some references, e.g. [34], are cited a number of times, whereas other references have fewer mentions.

 At times a study is cited in the paper by author's name without giving bibliographic reference, e.g., lines 232 and 325.

Section 6.2 on Modulation of Epileptic Network relies heavily on [34], however, it contains lesser contribution in terms of insight gained by the authors.

The statistical analysis of RNS data has not been covered in depth in this study except for improved seizure detection using PCA in [46]

Greater coverage of machine learning algorithms for seizure prediction is suggested and will enhance the value of this review paper.  

Though authors have mentioned the possibility and need for longer-term data collection using RNS [52], they did not mention the cyclical structure of epilepsy reported by [52].